# Mapping the learning curves of deep learning networks

**Yanru Jiang** [ID][1]*, **Rick Dale**[1]

Department of Communication, University of California, Los Angeles, Los Angeles, California, United States of America

* yanrujiang@g.ucla.edu

**Data availability statement:** All code used for running simulations, model fitting, plotting, and analysis is available in a GitHub repository at https://github.com/JoyceJiang73/Learning-Curves/. Data for conducting

## Abstract

There is an important challenge in systematically interpreting the internal representations of deep neural networks (DNNs). Existing techniques are often less effective for non-tabular tasks, or they primarily focus on qualitative, ad-hoc interpretations of models. In response, this study introduces a cognitive science-inspired, multi-dimensional quantification and visualization approach that captures two temporal dimensions of model learning: the "information-processing trajectory" and the "developmental trajectory." The former represents the influence of incoming signals on an agent's decision-making, while the latter conceptualizes the gradual improvement in an agent's performance throughout its lifespan. Tracking the learning curves of DNNs enables researchers to explicitly identify the model appropriateness of a given task, examine the properties of the underlying input signals, and assess the model's alignment (or lack thereof) with human learning experiences. To illustrate this method, we conducted 750 runs of simulations on two temporal tasks: gesture detection and sentence classification, showcasing its applicability across different types of deep learning tasks. Using four descriptive metrics to quantify the mapped learning curves—$start$, $end - start$, $max$, $t_{max}$—, we identified significant differences in learning patterns based on data sources and class distinctions (all $p$'s $<$ .0001), the prominent role of spatial semantics in gesture learning, and larger information gains in language learning. We highlight three key insights gained from mapping learning curves: $non - monotonic\ progress$, $pairwise\ comparisons$, and $domain\ distinctions$. We reflect on the theoretical implications of this method for cognitive processing, language models and representations from multiple modalities.

## Author summary

Deep learning networks, specifically recurrent neural networks (RNNs), are designed for processing incoming signals sequentially, making them intuitive computational systems for studying cognitive processing that involves dynamic contexts. There has

simulations is available from https://github.com/linuxsino/iMiGUE and https://huggingface.co/datasets/dair-ai/emotion.

**Funding:** The author(s) received no specific funding for this work.

**Competing interests:** The authors have declared that no competing interests exist.

been a tradition in the fields of machine learning and neuro-cognitive science to examine how a system (either humans or models) represents information through various computational and statistical techniques. Our study takes this one step further by devising a technique for examining the "learning curves" of deep learning networks utilizing the sequential representations as part of RNNs' architectures. Just as humans develop learning curves when solving problems, the introduced method captures both how incoming signals help improve decision-making and how a system's problem-solving abilities enhance when encountering the same situation multiple times throughout its lifespan. Our study selected two distinct tasks: gesture detection and emotion tweet classification, to illustrate the insights researchers can draw from mapping models' learning curves. The proposed method hinted that gesture learning experiences are smoother, while language learning relies on sudden knowledge gains during processing, corroborating the findings from previous literature.

## Introduction

Over the past decade, deep learning and neural networks have achieved remarkable performance in prediction and classification tasks in various domains, from machine translation and object recognition, to autonomous driving and reinforcement learning [1,2]. As powerful representational learning tools, deep neural networks (DNNs) can capture complex patterns in data [1], yet understanding the nature of the information embedded in their multidimensional representations remains a challenge. Researchers have raised concerns about how DNN embeddings represent knowledge and how to holistically interpret these high-dimensional features [3–6]. Over-reliance on such models for decision-making could be detrimental in both research and applied settings due to their complexity and lack of explainability. This is particularly concerning when models rely on biased training data that does not generalize well to target tasks [7,8]. Without a deep understanding of these models' underlying properties, they may fail to align with the goals of their human designers.

Recently, there has been a combined effort from cognitive science and deep learning to utilize representational learning to address model explainability concerns. These practices have become more prominent due to the success of large-scale models [17], especially large-language models [18]. For example, in the case of language models, examining the internal processes of the BERT Transformer-based architecture has shown that it may recapitulate common natural language processing (NLP) pipelines [19–21]. Chang and Bergen [22] found that the frequency and n-gram structure of word tokens significantly alters the training for language Transformer models learning these words.

Inspired by this prior work, this paper outlines a technique for examining the learning trajectories of deep learning models, in particular recurrent neural networks (RNNs). There is historical precedent for our approach, too. McClelland, Rogers and others have studied the underlying knowledge of neural networks by tracking them as they learn [10,23]. Despite this classic work in cognitive science, it is uncommon to see deep learning models that track progress as a way to unpack what is learned (e.g., going beyond simple RMSE curves; but see also [18], for a counterexample). We term this tracking a "learning curve," as it resembles research on how human learners process incoming information and improve decision-making through iterations under different situations. A unique benefit with simulation is the possibility to examine many dimensions and measurements of the neural network over time. In the next section, we review recent work on interpreting DNNs and related models. We then introduce our approach based on learning curves.

## Background

To date, various techniques have been proposed to interpret DNNs. For example, many model interpretability techniques provide task-specific and local explanations, such as saliency maps, attention maps, or Layer-wise Relevance Propagation (LRP), which interpret or visualize the localized influence of a region on the output. These ad-hoc approaches can be unstable, as even a minor change in a single pixel or hyperparameter can substantially affect the local relationships between input signals and output data [24]. Additionally, these local explanations fail to offer a global understanding of whether the selected architecture is well-suited to the task or how the DNN models experience learning. On the other hand, feature attribution methods like SHAP [66] and LIME [67], while conceptually easier to interpret and model-agnostic, tend to work more intuitively with interpretable dimensions typically found in structured tabular data. These methods are less effective for unstructured tasks, such as image and audio data, where feature dimensions are harder to define or interpret. While most techniques emphasize the qualitative interpretation of a model, such as what has been learned or captured by the DNN, the absence of quantifiable measurements makes cross-model comparisons in unstructured tasks particularly challenging [4,5,25]. We further provided a systematic description of different model explainability methods, along with their pros, cons, and use cases, as well as a comparison to our proposed approach in Table 1.

In a foundational article on deep learning, LeCun and the colleagues [1] characterized DNNs as representation-learning methods utilizing multilayered large neural network-style models. Representations can be viewed as mental objects capturing semantic properties either observable or unobservable [26]. Different from traditional machine learning models, DNNs display remarkable flexibility and efficiency in encoding lower-level input signals, such as pixels, audio frequencies, or word tokens, into multidimensional vectors at a sophisticated level through multilayered nonlinear transformations [27]. These transformations generate multiple levels of representations that learn hierarchies of features at each layer [1]. The continuous numerical vectors (or hidden vectors) learned at each level are commonly referred to as "embeddings" and serve as dense representations of the original input data. Following this construction, numerous studies have demonstrated the correspondence between

**Table 1. DNN Explainability Methods Comparison.**

| DNN Explainability Method | Description | Advantages | Limitations | Use Cases |
|---|---|---|---|---|
| Learning Curve (Proposed Method) | Track the model's learning experience over time by utilizing classification performance of embeddings. | Captures dynamic changes in embeddings during training. Insight into performance evolution across two temporal dimensions. | Limited to temporal tasks and DNN architectures. Computationally expensive. | Tasks involving temporal aspects and learning experiences. |
| Saliency Maps | Visualizes regions in image input that most influence the output. | Simple and intuitive visual explanation. Easy to implement. | Prone to noise. Lacks global interpretation. | Image recognition tasks and image-based models. |
| Feature Attribution | Infers relevance of input features to output. | Conceptually easy to grasp. Model-agnostic approach. | More suited for structured data. Less intuitive for unstructured data. | Tabular data and structured tasks. |
| Layer-wise Relevance Propagation (LRP) | Breaks down classification decision into contributions of input elements. | Applicable to a wide variety of DNN models. | Lacks global interpretability. Requires complex configuration. | Image recognition, NLP, structured data. |

DNN-generated and real-world distributed representations among words and sentences [11, 28], speeches [29], images [9], objects [30] and scenes [31,32]. Representational learning in DNNs offers fundamental contributions to cognitive science, as it can inform how cognitive systems process and organize knowledge, facilitating the comparison of learning processes between humans and machines [33–35].

Recently, several studies in cognitive science and neuroscience have highlighted the importance of integrative modeling between computation, human brains and behaviors [12]. Beyond comparisons of static end-point knowledge, scholars have begun exploring the potential correspondence in learning and information processing between DNNs and human cognitive systems, given that the current performance of DNNs can already approximate human performance across various domains [36–38] (see [18] for a review). For instance, the representations (i.e., embeddings) extracted from multilayered DNNs have shown significant accuracy in predicting neural and behavioral responses in humans throughout the hierarchy of learning and processing. This evidence spans multiple neural-behavioral measurements (e.g., fMRI, EEG, ECoG), modalities (e.g., visual, auditory, and language processing), and model architectures (ranging from simple embedding models like GloVe to more complex neural networks such as RNNs, convolutional neural networks (CNNs), and transformer models (for further details, see [9,11,12,23,39,40]).

Given the extensive alignment observed between DNN embeddings and neural-behavioral activities, and their presumed meaningful representational mapping with human cognitive systems, an analysis of how they emerge in learning would seem important to understand these relationships. Goldstein and colleagues [11] identify the temporal correspondence between layer-by-layer embeddings in GPT-2 and evolving neural activities in language areas. However, this temporality is restricted to layerwise representations (from low-level to high-level representations) rather than how streams of signals have been received and processed by models or brains [38]. Our aim in this paper is to use temporal analysis in a systematic way by separating and tracking the time course of a network's learning across classification tasks, thereby enhancing the understanding of the emergence of representations in DNNs.

## The current study

The goal of this study is to unpack the "learning curve" of DNNs through a sequence of hidden representations when the model encounters any temporal processing tasks across its training. To do so, we sample the network's performance by using its embedding vectors to classify groups of items in its training input. This allows us to map out the progression of the network's discriminations across these groups – how the network's internal knowledge, in the form of embedding vectors, evolves during training.

The model architecture we focus on is RNNs due to their capacity to model sequential data and time-dependent tasks [41], such as text generation, speech recognition and stock market prediction. Although other deep learning architectures, such as CNNs and Transformers, also have the capacity to process time-series data, RNNs are explicitly designed for processing sequential data, as they effectively capture temporal dependencies through their recurrent connections. CNNs excel in tasks such as image recognition by applying the kernel trick, where convolutional filters are used to extract spatial features from local regions of the input. While CNNs can be adapted to handle sequential data using techniques like 1D convolutional layers, they lack the inherent ability to capture temporal ordering in the data, as they process each segment of input independently. Transformers, on the other hand, rely

on self-attention mechanisms to weigh the importance of different input tokens in parallel, allowing them to capture dependencies between distant elements (i.e., tokens). This feature makes Transformers better suited for tasks where relationships between tokens are not strictly ordered in time, and their parallel processing nature is less ideal for learning tasks that require explicit temporal progression. In sum, the strict sequential processing nature of RNNs makes them an intuitive architecture for studying cognitive processing that involves dynamic, changing contexts [42].

In this study, we propose a generalizable interpretability approach that maps the global learning curves of RNNs based on changes in classification performance between embeddings and output across timesteps of the input data. This performance reflects the predictive capacity or the amount of signal captured by the embeddings in predicting the final output. In RNN modeling, each timestep generates an embedding (e.g., in language processing, each timestep represents a word). These embeddings progressively incorporate information from the beginning up to timestep $t$. By examining changes in classification performance between these embeddings and the output, we can capture how the information evolves over time. To provide clarity, we use the term "learning curve" in this study to denote the holistic approach and intention behind mapping the underlying processing and developmental journey of DNNs. The method we propose separates two parts of the learning curve, one based on overall training, and another based on processing within input items during training. First, the "developmental trajectory" signifies the long-term learning process of DNNs, which is simulated by the increasing number of epochs (i.e., complete passes through an entire training dataset). Second, the "(information) processing trajectory" refers to the momentary accumulation of information across all timesteps (within an epoch), extractable from the performance of the RNN layer (see Fig 1 for a conceptual illustration). We detail each of these further below.

As Fig 1 illustrates, the resulting visualization consists of learning curves with timesteps on the x-axis and performance on the y-axis. Each epoch contains an information processing trajectory, and the set of curves together forms a developmental trajectory. Due to the multidimensional nature of this plot, while it provides rich qualitative information about the entire model learning experience, it can be challenging for researchers to comprehend all the constructed curves at once. Therefore, beyond the visual presentation and qualitative inspection of the multi-dimensional learning curves, this study further defines four measures: start performance (*start*, the initial capacity of each information processing), max performance (*max*, the maximum performance of each information processing), time at max ($t_{max}$, when the current information processing reaches the maximum performance), and end – start performance (*end – start*, the overall performance gain in this information processing) – to facilitate quantitative comparisons across tasks and datasets. Our study particularly focuses on these relatively easy-to-comprehend descriptive statistics (as opposed to more complex metrics like regression coefficients) to simplify the understanding of the already multidimensional and granular nature of the constructed learning curves.

To illustrate the generalizability of our approach, we chose two distinct classification tasks: (i) sentence classification and (ii) gesture detection. These tasks differ widely in terms of their modalities. We predicted this would lead to variation in the underlying data generation processes and associated cognitive processing for each modality. In particular, gesture and body movements primarily result from the coordinated contraction and relaxation of muscles, with signals produced at later timesteps derived from the previous timesteps with relatively high autocorrelation [43]. On the other hand, verbal language, being a predominantly semantic modality, exhibits degrees of surprisal and arbitrariness that enhance the cognitive capacity

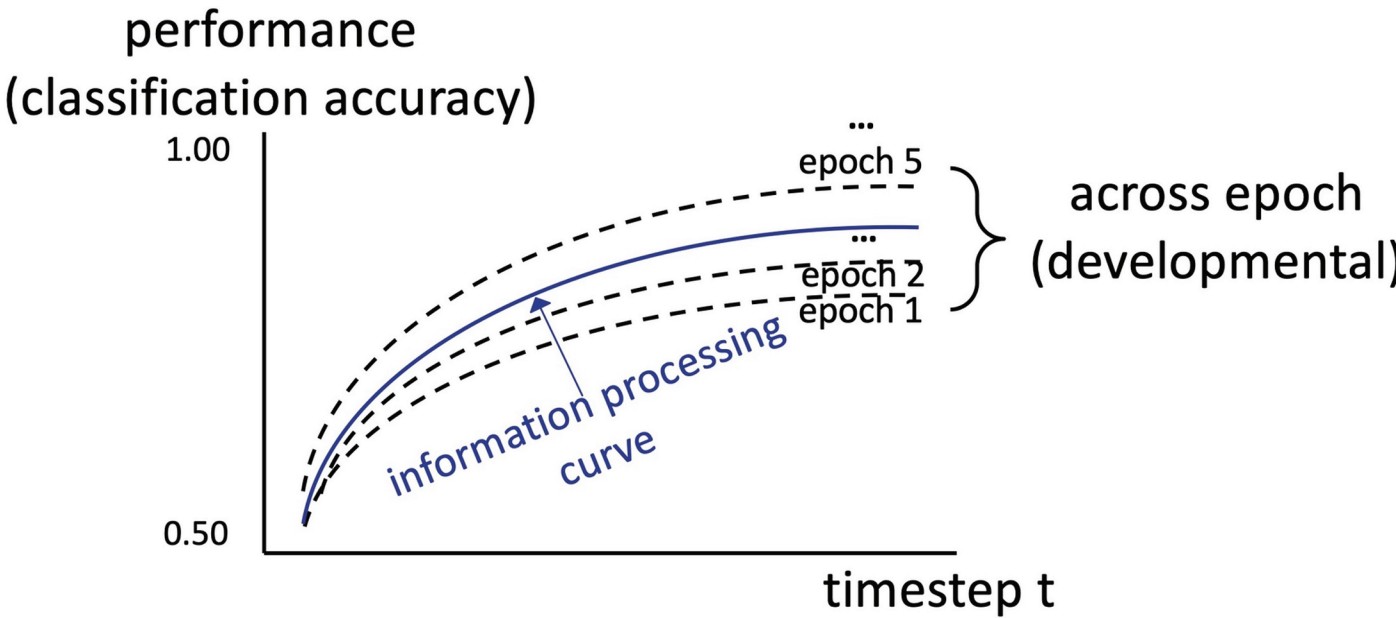

**Fig 1. A conceptual illustration of learning curves in DNNs.** The solid curve represents an individual information processing trajectory (across timesteps). For example, as an agent receives more signals over time in one session, its prediction of the gesture increases. The bundle of dotted lines represents a developmental trajectory (across epochs). For instance, this agent improves its ability to predict incoming gestures after repeatedly encountering similar patterns. In a typical binary classification task, the starting point at $t=0$ is expected to be near 0.50 (and gradually increase to 1.00 across timesteps and epochs) for all epochs because no signals have been provided at the first timestep for solving the underlying task. Hence, all information processing curves in this figure have the same starting point to reflect this pattern. In practice, certain classification tasks might contain structural information even at $t=0$ (e.g., gesture classification contains spatial information, such as the location of keypoints, at the initial timestep, which could help solve the gesture task from the very beginning).

of language processing [44–46]. Therefore, the expected developmental and information processing trajectories will likely exhibit distinguishable patterns across the two different tasks when the DNN system processes them respectively.

As we discuss in detail in later sections, this method has a few benefits. Through tracking the learning trajectory of a neural network, researchers can explicitly identify the appropriateness of a model for a given task as well as examine the properties of underlying input signals. This approach could also serve as a standalone visualization to map the accumulation of the underlying signals processed, which can facilitate research on deep learning modeling and signal processing across various modalities. Finally, mapping the learning curve of DNNs has the potential to assist future computational cognitive and neuroscience research and address whether the learning experiences of models also correspond to (or fail to correspond to) the temporal processing in human cognition in addition to the emphasis on static representations in the current literature.

In the following sections, we will provide a step-by-step method for visualizing the learning curves of neural networks, illustrate how to holistically interpret signal processing in them and quantitatively compare these curves across two different datasets.

## Methods

This research proposes a model-interpretability method that can extract the learning curve of sequence-based deep learning networks (e.g., RNNs). Inspired by cognitive science, the

method measures the learning trajectory and underlying knowledge extracted by such networks. To illustrate the method, we use two temporal tasks: gesture detection and sentence classification as examples. This study therefore demonstrates that the method could inform a range of deep-learning tasks.

Our multi-stage pipeline includes three main steps. First, we trained the RNN-LSTM model to generate a sequence of embeddings for a temporal task. Next, we used KNN classifiers to calculate the performance across all extracted embeddings, which we refer to as learning curves (see (2) in Fig 2). Finally, we defined four metrics to quantify these multi-dimensional embeddings for more systematic comparison and significant tests between tasks, classes, the null hypothesis, and development trajectories. The visualization of this multi-stage procedure can be found in Fig 2.

## Datasets

This study utilized two datasets: the Identity-free Video Dataset for Micro-Gesture Understanding and Emotion Analysis (iMiGUE) by [48] and the Emotion dataset from Hugging Face by [49] to examine the possibility of mapping learning curves for tasks involving temporality. The two temporal tasks (i.e., gesture detection and sentence classification) are distinct in terms of their modalities, lengths, and the steps required to extract and preprocess the features, thereby enhancing the diversity of data to illustrate the learning curve analysis. We detail the preparation of each dataset separately below.

**Emotion tweet classification (sentence classification).** The Emotion dataset [49] consists of 20,000 English Twitter messages with six basic emotions (e.g., anger, fear, joy, love, sadness, and surprise) by adopting Plutchik's [50] wheel of emotions, Ekman's [51] six basic emotions,

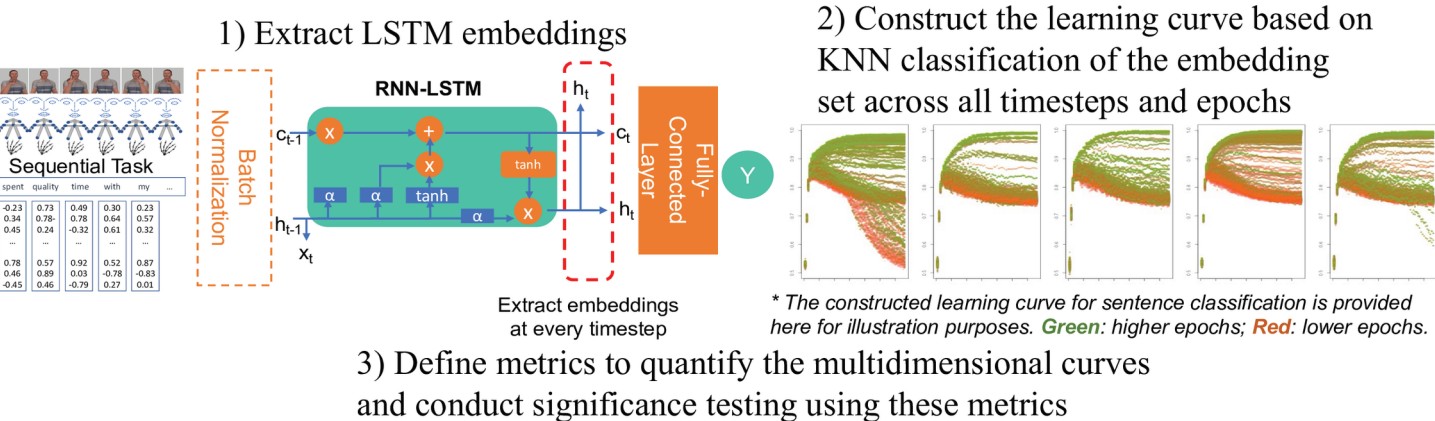

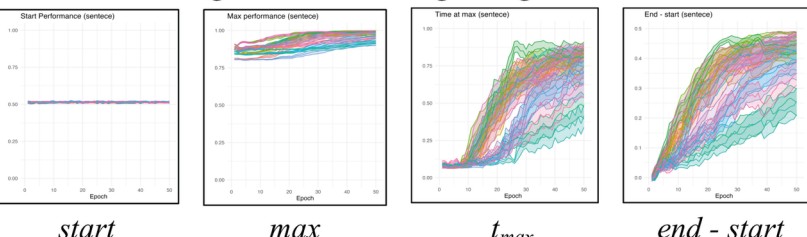

**Fig 2. A flowchart of the multi-stage pipeline.** Illustrations of the multi-stage pipeline, from extracting embeddings to constructing learning curves and quantifying the curves using four defined metrics. NB: Gesture figures are adapted from Wikimedia Commons [47].

and hashtags in tweets. Tweets were annotated through noisy labels and distant supervision introduced by [52].

To prepare the emotion sentences for recognition by the RNN layer, we first applied the "basic English" tokenizer from torchtext to tokenize each tweet. Then, we used GloVe (Global Vectors for Word Representation), a word vectorization technique that does not rely on local word context statistics (local-context information), to vectorize each token in a sentence. GloVe was preferred over other token vectorization techniques like word2vec [53] due to its design to capture the universal meaning of each token/word, rather than the word's meaning within a specific sentence or context. We opted for GloVe embeddings with 300-dimensional semantic features because it strikes a balance between capturing sufficient information and maintaining computational efficiency [35].

In theory, the input sequences of RNN are not required to have the same length. In practice, these sequences are padded with zeros or trimmed to the same length to optimize the computation in PyTorch. Accordingly, all tokenized tweets were padded to a consistent length of 66 tokens, which corresponds to the length of the longest tweet sample. The shape of each Emotion tweet follows a 300 dimensions × 66 timesteps (see Fig 3).

**Gesture classification.** The iMiGUE is a high-quality dataset that contains 18,499 identity-free, ethnically-diverse, gender-balanced samples of 32 psychologically-meaningful micro-gestures (such as scratching an arm, adjusting the hair, or touching an ear [51]). These gestures were all curated from interview clips with athletes at post-match press conferences. Unlike other gesture and emotion datasets, which are typically drawn from staged performances or movie clips, the iMiGUE provides samples of actual gestures from real-life situations. This poses a more realistic, though more challenging, recognition task for deep learning networks [54,55].

Due to copyright restrictions, the dataset includes only skeleton keypoints (rather than original interview clips) extracted from OpenPose, a multi-person computer-vision system that can simultaneously extract keypoints of the body, hands, face, and feet [56][1]. In total, 25 body, 70 facial, and 21 left hand and 21 right hand keypoints were extracted for each frame using OpenPose. Keypoint data were stored in the following format: [x0,y0,c0,x1,y1,c1… ], in which (x, y) represents the coordinate of each keypoint and c indicates the confidence score of each keypoint-coordinate prediction. The confidence score was excluded in the following processing steps.

Although the gesture clips have just 39 frames on average, they have a higher standard deviation at 84 frames with a 75th percentile of 72 frames. We therefore set the padded length of the input sequence to 150 frames to ensure that our input data contained sufficient information for training a classifier. Thus, the input size for each gesture clip was 274 units (137 keypoints × 2) × 150 timesteps (see Fig 3).

It is important to note that our aim here is not to approach benchmark performance, but rather to examine successful learning. The learning curve analysis will show how that successful learning emerges, and which stimulus discriminations seem to underlie that emergence. Since even the state-of-the-art neural network can achieve only 55% accuracy on this multi-classification task [48], we further grouped the 32 micro-gestures into six categories (body, head, hand, body-head, head-hand movements and an absence of gestures) to ensure that our RNN-LSTM was indeed "learning" when we attempted to map its learning curve.

---

[1] The advantages of using key point data are twofold. First, keypoint data conserves significant computing power; since each second of image sequences (i.e., matrices of pixels) may contain as many as 30 or 40 frames, running deep learning models on image sequences can be prohibitively expensive. Second, it provides better interpretability and understandability. Instead of being possible only through abstract information at the image-frame level, gesture detection can be operationalized as a sequential movement in keypoints across frames [57].

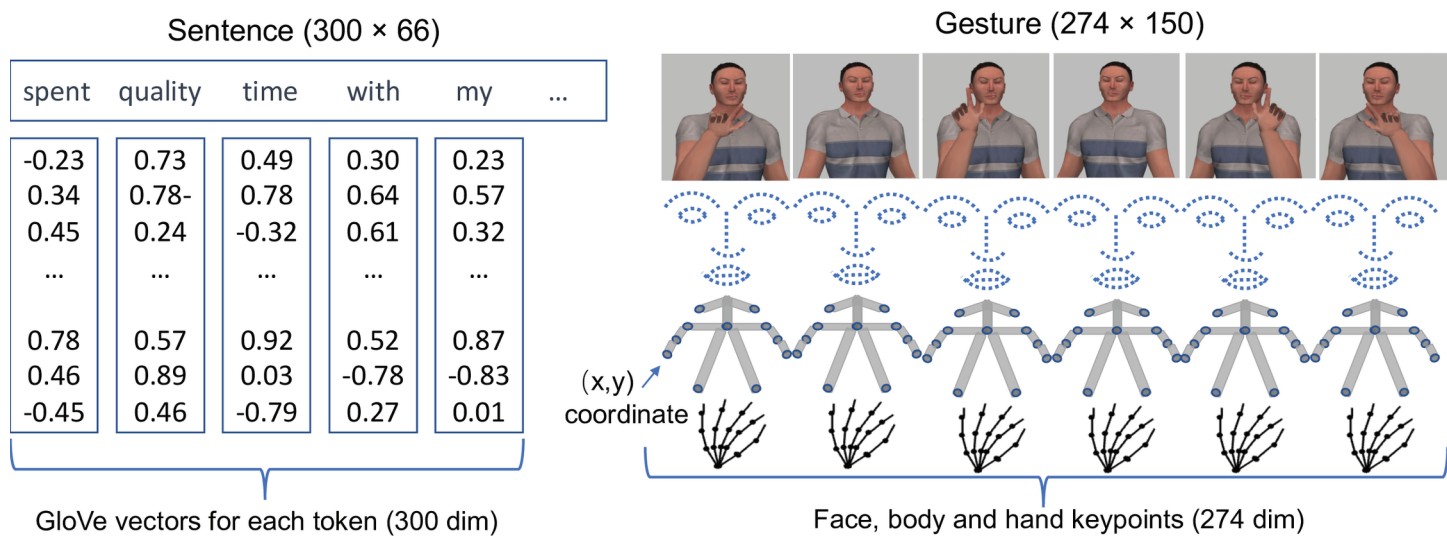

**Fig 3. Sequential data for Emotion Tweet (left) and Gesture Classifications (right).** NB: Gesture figures are adapted from Wikimedia Commons [47].

## Model architecture

We selected a model architecture (see Fig 4) that is standardized for sequential stimulus processing in the following way. First, we had a batch normalization layer, a primary RNN-LSTM layer, and then a fully connected layer to connect the final dense embeddings with the output classes (for classification). We offer details below.

 **RNN-LSTM.** RNN is capable of modeling sequential data and time-dependent tasks [41]. Its architecture represents an iterative function that takes an input sequence ($x$) and

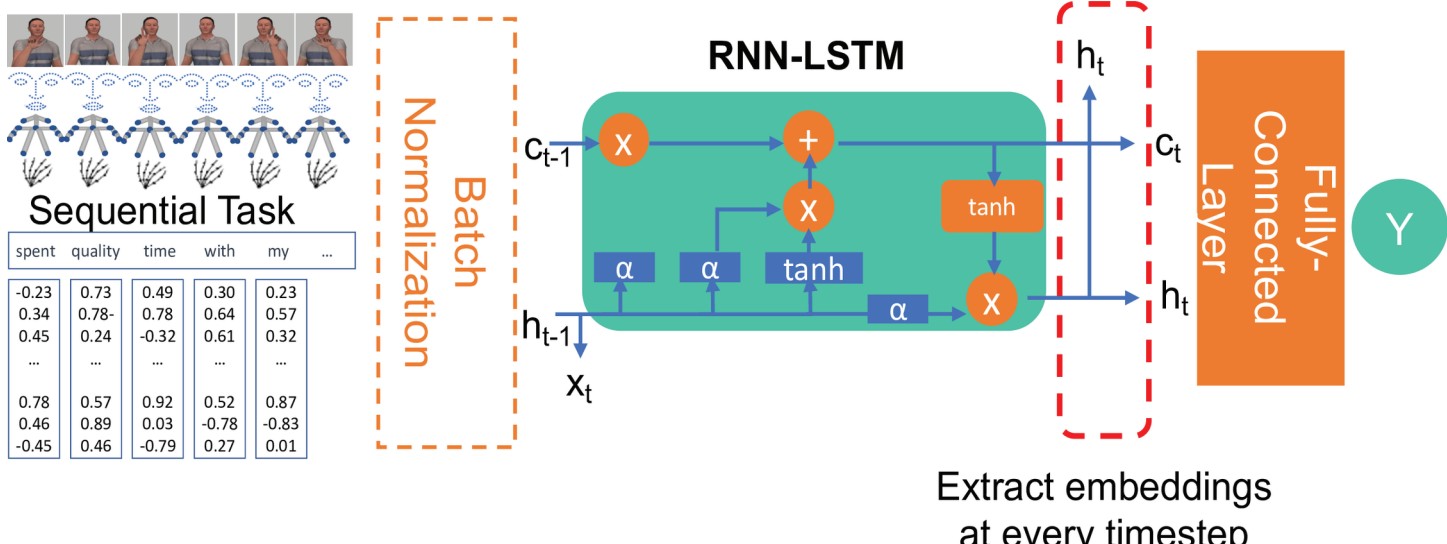

**Fig 4. Model architecture of RNN-LSTM.** NB: Gesture figures are adapted from Wikimedia Commons [47].

an internal state ($h$) from the previous timestep ($t$ - $1$) to predict the current timestep ($t$), then updates the state as follows:

$$h_t = f(x_{t-1}, h_{t-1}) \quad t = \{0, 1, 2, \dots, T - 1\} \tag{1}$$

As the formula illustrates, each timestep $t$ should theoretically reflect the information from 0 to $t - 1$. We selected the RNN model for gesture detection because it can process temporal information under the assumption that the body movement in each timestep depends on signals in the previous timesteps.

While an RNN could leverage the context between elements by maintaining its internal state while processing the entire sequence, the "Vanilla RNN" layer experienced the vanishing-gradient problem during model training [58]. Therefore, long short-term memory (LSTM), which is represented by the function f in Equation 1, was introduced here as an additional state variable, called the cell state, for controlling specific information that needed to be kept or updated while processing the entire sequence [57]. LSTM effectively reduced the vanishing-gradient problem encountered by RNN [58].

**Feedforward network.** The construction of our RNN-LSTM neural network follows common practice in deep learning. First, to enhance training stability and speed up convergence, we applied batch normalization, specifically the BatchNorm1d layer, to the input data at each timestep, with dimensions batch_size × timestep × input_dimension ($1 \times 150 \times 274$ for iMiGUE and $1 \times 66 \times 300$ for Emotion). This means that each timestep is treated independently, and normalization is applied across all data points in the batch for each timestep. Specifically, this technique normalizes the inputs to have a mean of 0 and a variance of 1 within each timestep, standardizing the distribution of inputs across the data points in each batch during training. Then, an RNN-LSTM was applied to the normalized input to convert temporal information to a dense embedding at each timestep. An RNN-LSTM with an equal hidden dimension was selected to simplify the tracking of embeddings in the later stage. Finally, a fully connected layer, without any activation, was applied to the embeddings in the last timestep $t$ to predict output labels.

## Deep learning simulation

To ensure the generalizability of our learning curve mapping approach, we performed 15 rounds of simulation for both gesture and emotion detection tasks, each of which included 15 (6C2) pairwise binary classifications. We conducted 25 repetitions (reps) for each set of simulations (sims) to mitigate the idiosyncrasies of specific processing and developmental patterns we are extracting under each pairwise condition. In total, we collected $25 \times 15 \times 2$ ( reps × sims × tasks) = 750 runs of simulation data. Having an adequate number of simulation runs also enables us to observe clustering tendencies in the convergence and divergence of trajectory patterns across various tasks and different classes.

In each simulation, 20% of the shuffled samples were used as the test data, while the remaining data were further split into 80% training data and 20% validation data. While validation data were used for reporting each epoch's model performance, test data were used for reporting the final model performance on unseen data.

The number of epochs indicates the number of times an entire dataset has been passed forward and backward through the neural network. We set the number of epochs to 50 for both tasks, given that most of the variation in learning tends to unfold during the early stages of development. In S3 Text and S4 Text we also illustrate how using 20 and 100 epochs leads to similar observed patterns.

Because deep learning algorithms are very sensitive to unbalanced datasets, we applied data augmentation to the training and validation datasets. Specifically, the minority class was up-sampled to match the number of data points in the majority class, which ensured a balanced dataset [59].

Default initiation from PyTorch was used to standardize the model specification across simulations. All simulation runs were trained using the Adam optimizer with a learning rate of 0.001, and the loss function used was the cross-entropy loss for all pairwise classifications. All the other learning rate hyperparameters were kept at their default values.

Since deep learning is, computationally speaking, very expensive to train, the model was run on Nvidia RTX 4090 to expedite the processing. We set the batch size to 64 for gesture recognition and 256 for emotion classification to achieve an optimal balance between training speed and performance.[2]

To map a learning curve for each simulation, we first extracted embeddings of all LSTM timesteps from the hidden layer. We then applied multiple interpretable machine-learning models between these embeddings and the corresponding output labels to understand how the model's confidence is updated during the LSTM timesteps.

**Embedding extraction.**   We extracted the LSTM array of all timesteps for all batches across all epochs on the test data to ensure that we were extracting embeddings from all developmental stages and the final model was adequately trained on the targeted task. Since we specified the same dimension of the input data for the LSTM hidden layer, for the gesture detection task, we obtained 150 LSTM arrays from 150 timesteps, each array having the size of $1 \times$ hidden_dimension ($1 \times 274$). Similarly, we obtained 66 LSTM arrays in sentence classification, each with a size of $1 \times 300$. We then stored the corresponding output labels of those arrays as an output array. Those LSTM arrays share one output array since they are embeddings at different timesteps of the same data points.

For binary classification, the labels were encoded as 0 and 1. Once the model had been trained and the embeddings extracted, we stored and processed those embeddings in RAM, which has a greater storing capacity than GPU.

**Mapping the learning curves.**   To approximate the processing curve of a model's confidence across the LSTM timesteps, this study used a popular machine learning algorithm, k-nearest neighbors (KNN) for identifying embedding "separatables." This algorithm classifies an object by using a majority vote of its neighbor data points [60,61]. Because embeddings are seen as a high-dimensional physical (i.e., location-wise) projection of input data (as opposed to a multivariate representation [1]), distance-based models, such as KNN and support vector machine (SVM), are popular choices for examining DNN embeddings in previous studies. Specifically, we applied KNN to the data point $(x_t, y)$, where $x_t$ is the LSTM embedding at $t$ timestep and $y$ is the corresponding output label and calculated the KNN accuracy across all timesteps (information processing trajectory) and all epochs (developmental trajectory), and thus captured the learning curve of the model.

## Quantifying performance across epochs

These learning curves examine how the LSTM's embeddings classify the stimulus as it is incrementally presented to the network. Visually, each curve is plotted as the proportion of correct classifications across the item. These constructed learning curves can be analyzed

---

[2]   The large batch size for emotion classification is due to each of its input data points using less space compared to the gesture input.

in various ways, either through qualitative interpretation of the visualizations or by applying statistical analysis for quantification and hypothesis testing. In this study, we utilized the four descriptives statistics introduced earlier to assess the multidimensional KKN classification performance of the RNN-LSTM (i.e, how the discriminability of the model is evolving) across its training on two datasets, reconstructing and analyzing the unfolding learning dynamics of these distinct learning tasks. Specifically, for each such curve, we extracted four simple descriptive metrics: *start*, *end - start*, *max*, $t_{max}$. In Fig 5, we illustrate these descriptive statistics from an example trajectory of a single classification run for each dataset.

First, we defined the maximum performance (*max*) and the percentage of time at which that maximum was achieved in the presented item ($t_{max}$) over the whole timesteps. We

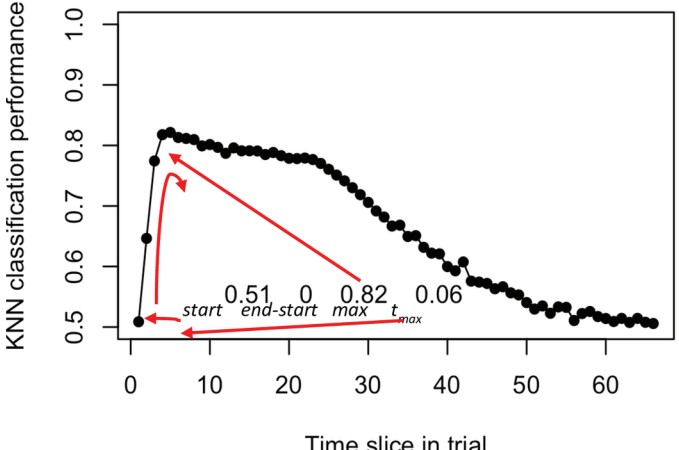

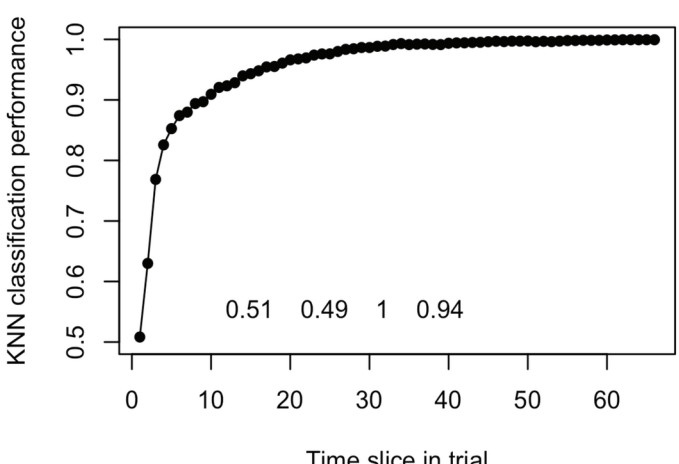

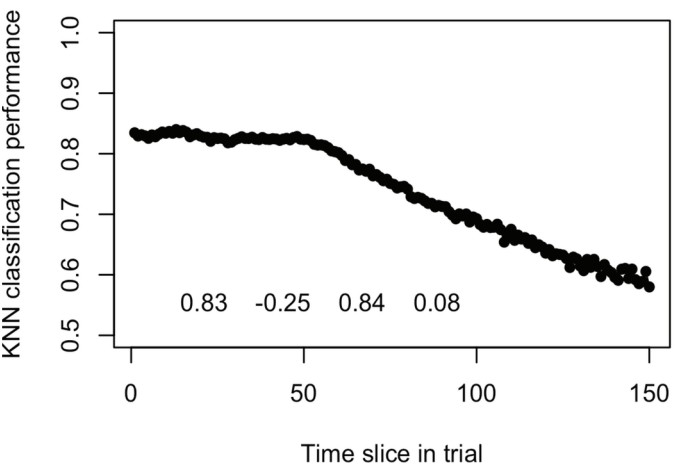

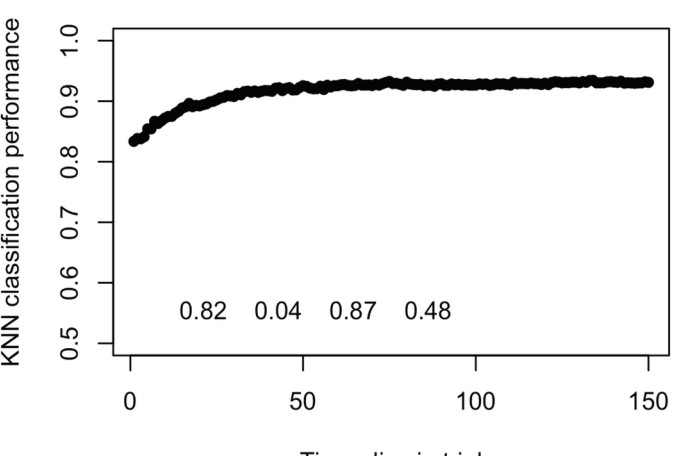

**Fig 5. Examples at epoch 1 and 50 of binary classification across a test input for one run of the network.** From epoch 1 to epoch 50, the LSTM's embeddings are able to classify successfully over the test item, and we can characterize this success as a change to its performance using four metrics described in the main body of the text (*start*, *end - start*, *max*, $t_{max}$). Top: Example item from the sentence task. Bottom: Example item from the gesture task.

included $t_{max}$ alongside *max* because we discovered that *max* alone is not sufficient to determine performance, especially earlier in training. This is because the network may exhibit unstable performance, dropping across subsequent time slices, which may be indicative of a non-monotonic learning trend seen in related domains [62]. This also suggests the network has learned something about the *initial* segments of an item, but the later segments wash out its performance as it has not yet encoded these later features. To capture this trend, we use $t_{max}$ to assess when that maximum was achieved within each information processing trajectory – specifically, the percentage of the time slice at which the observed maximum occurred. In Fig 5 above, we show an illustration of this in the top left. The network achieved a performance of 0.82 on this particular item, but failed to sustain this performance as it dropped to near 0.50 as the sequence unfolds (at epoch 1). In the top right, the performance improves approximately monotonically across time slices (by epoch 50), and $t_{max}$ is achieved near the final bin of the training item.

Additionally, we assess the performance at the start and end of the presented stimulus (*start*, *end*). Performance at the *start* may indicate the relative gains that can be expected from a stimulus item. As shown in the bottom panels of Fig 5, the gesture input already has performance above chance ($\sim 0.80$) after the very first segment of the stimulus item. This suggests the model rapidly exploits spatial information in gesture (i.e., the location of keypoint coordinates can already vaguely differentiate gesture classes before sequential movement information is fed into the model). A model that achieves near-perfect performance at *start* and sustains it does not need to be exposed to the subsequent stimulus. The last measure we use is the subtraction of *end - start* performance. A high value on this measure suggests the network gets substantial information gains across a test item. For example, even at epoch 50 in the bottom right, the *end - start* of the gesture item is substantially lower than the simulation trained on the sentence task.

These measures can be plotted across epochs. Each learning curve now indicates how an item is being processed across an LSTM's overall training. The measures are relevant to two timescales in the network's behavior we conceptualized earlier: developmental (or learning) and information-processing timescales. For example, across epochs, movement along the *start* measure represents the initial performance at the item's first time slice at the beginning of each information processing session. The *end - start* measure within an epoch can describe the relative information gains during each processing session from the item's full presentation. The *max* represents the best performance achievable in a session, indicating the network's current information processing capacity. Finally, if $t_{max}$ is low, it suggests that the network's maximum performance of the current session is hindered by subsequent timesteps, implying that additional training may be necessary.

## Results

We took the output from the KNN classification in Python, described above, and designed a sequence of R scripts to measure, visualize and quantify the trends in these four measures (*start*, *end - start*, *max*, $t_{max}$). R's suite of visualization tools provided a convenient arena within which to view trends across epochs, and in these analysis scripts we also built linear models to statistically test the significance of these trends. All of the scripts in our methods are available at GitHub here: https://github.com/JoyceJiang73/Learning-Curves/. The R scripts only require input of simulation CSV data that contain as fields: binary classification labels, time slice, epoch, and performance measure (e.g., KNN classification performance).

We have reported the trained RNN model performance for the average of all pairwise simulations of sentence and gesture classification (epochs = 50) in S1 Text, including metrics

such as validation accuracy, validation loss, test accuracy, test loss, recall, precision, and F1. These results confirm that almost all pairwise classifications have been sufficiently trained, indicating that the learning curves are capturing the learned experiences.

For each of the four metrics, we conducted a series of significance tests using regression models to assess distinctness across data sources, reshuffled null data, classes, and epochs. Specifically: (1) A linear model was used to assess whether the learning curves of the two datasets differed significantly, (2) A linear model evaluated whether the learning curve for each training classification task significantly deviated from a superimposed (randomly reshuffled) null hypothesis across the four metrics, (3) Linear models were used to assess whether the learning curves for each dataset were distinct by class and across epochs, and (4) An ANOVA test was performed to determine whether class differences contributed more to developmental progression than epoch alone. All the significance tests (Adjusted R-squared) are reported in Table 2, and the corresponding visualizations for each metric are provided in their respective sections.

## Start performance

In Fig 6, we show the LSTM performance at the *start* of an item across epochs of training. In general, the sentence dataset shows low, near-chance performance, while gesture classifications are already well above chance performance. This chance-level performance for sentence items stays consistent across the entire training period, though the gestural dataset shows some improvement. For the gesture dataset, this suggests that the network has some information about a classification before much of a training item is even shown to the network. It would indicate that gestural data has spatial information in the point coordinates of the body and is exploited by the network at the very first time bin. With language, it takes time for word embedding vectors to be integrated in the network.

To confirm these trends, we tested a linear model that predicted *start* performance by training data, showing that dataset accounted for about 98.14% ($p < .0001$) of the variance seen in Fig 6. The classification of the gesture data accounts for 95.51% ($p < .0001$) of variance

**Table 2.** A series of significance tests using regression models to assess distinctness across data sources, reshuffled null data, classes, and epochs.

| Model | start | max | $t_{max}$ | end-start |
|---|---|---|---|---|
| **Data Source Difference** | | | | |
| Data Source | 0.9814**** | 0.0211**** | 0.0034**** | 0.3923**** |
| **Baseline Null** | | | | |
| Sentence Reshuffle | 0.0001 | 0.0042**** | 0.0120**** | 0.0027**** |
| Gesture Reshuffle | 0.0824**** | 0.0722**** | 0.0058**** | 0.0340**** |
| **Epoch and Class Differences** | | | | |
| Sentence Class | 0.1022**** | 0.2736**** | 0.0951**** | 0.1301**** |
| Gesture Class | 0.9231**** | 0.8250**** | 0.2495**** | 0.3808**** |
| Sentence Epoch*Class | 0.1028**** | 0.6215**** | 0.4910**** | 0.5232**** |
| Gesture Epoch*Class | 0.9551**** | 0.9732**** | 0.6315**** | 0.6530**** |
| **Contribution of Class Differences to Epoch (Developmental) Progression** | | | | |
| Sentence Epoch*Class - Epoch | 0.1045**** | 0.3010**** | 0.1160**** | 0.1437**** |
| Gesture Epoch*Class - Epoch | 0.9336**** | 0.8539**** | 0.2868**** | 0.4282**** |

*Significance tests using regression models to assess distinctness across data sources, reshuffled null data, classes, and epochs for the four metrics (*start, end - start, max, $t_{max}$*). Adjusted $R^2$ values are reported along with the significance levels (* $p < 0.05$, ** $p < 0.01$, *** $p < 0.001$, **** $p < 0.0001$).

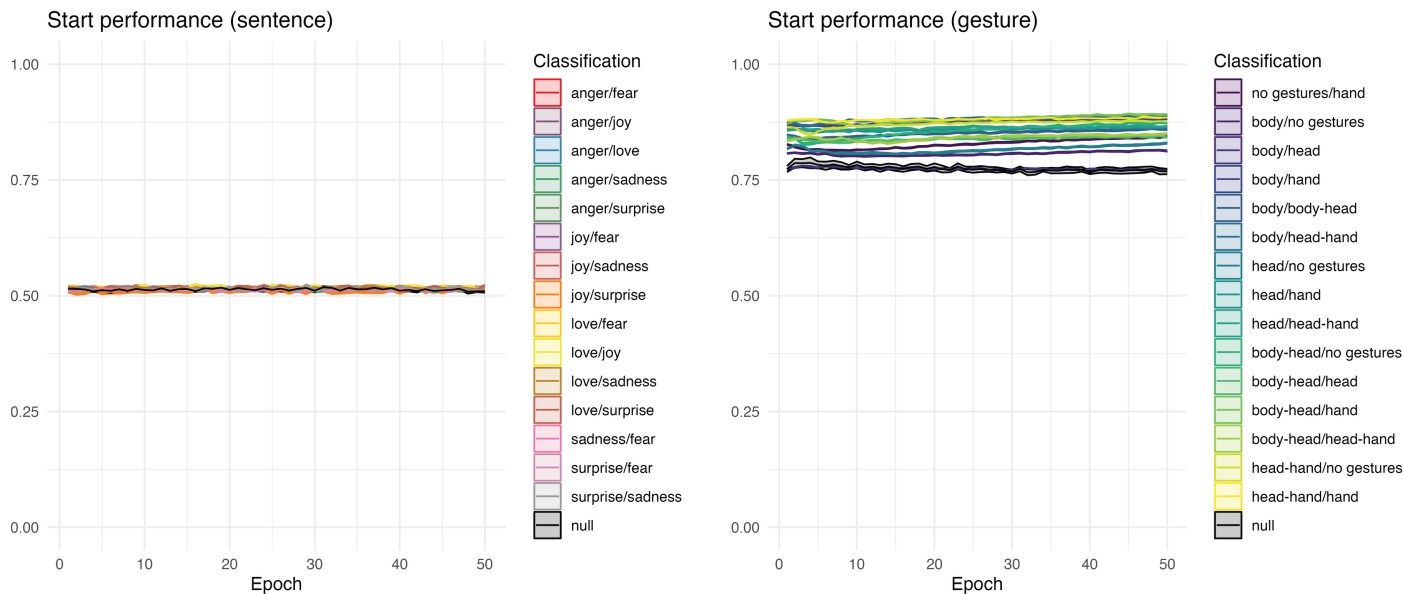

**Fig 6. *Start* performance for pairwise classifications.** *Start* metric over epochs, indicating that in the sentence task, the classification at the first time slice remains stable near change, whereas the LSTM's embeddings for the gesture task are distinct across stimulus types, and also show slightly more improvement over training (while already being well above change relative to the sentence task).

internally to that dataset, whereas for sentence classifications accounts only for 10.28% (lower but also significant, $p < .0001$). The learning curves reveal that the first word of a language task has low diagnostic accuracy for a classification, but the spatial variance over gestural classifications is much more informative.

The *start* performance for sentence null is the only metric that shows no significant difference between learning from ordered sentence classes and reshuffled data (among all significance tests; $R^2 = 0.0001$, $p < .0879$). This is expected, as all sentence binary classifications begin with a consistent initial performance of 0.50 (random guessing for binary classification), which remains unchanged even with randomly reshuffled sentence data. In contrast, the *start* performance significantly differs for ordered gesture classes versus reshuffled data because the various gesture pairs contain different levels of spatial information ($R^2 = 0.0824$, $p < .0001$), affecting their initial performance.

## Max performance

Curiously, if one only investigated maximum performance across training, these classification tasks could be regarded as relatively similar in their behavior. As shown in Fig 7, both sentence and gesture datasets yield a classification performance that is high, between 0.75 and 1.0 depending on the classification. In the gesture dataset, there are more "difficult" classifications, shown by outliers in *max* performance across training. This can be helpful in diagnosing representational challenges in the network's training, marking what pairs of training items may be more difficult to distinguish than others.

Again, as with the *start* measure, the *max* performance shows greater variance associated with classifications in the gesture case ($R^2 = 0.8250$, $p < .0001$) than the sentence case ($R^2 = 0.2736$, $p < .0001$). The difference between these two datasets is not as pronounced as in the *start* measure, as a linear model shows that only 2.11% ($p < .0001$) of the variance is associated with the dataset in a linear model predicting *max* performance observed within a

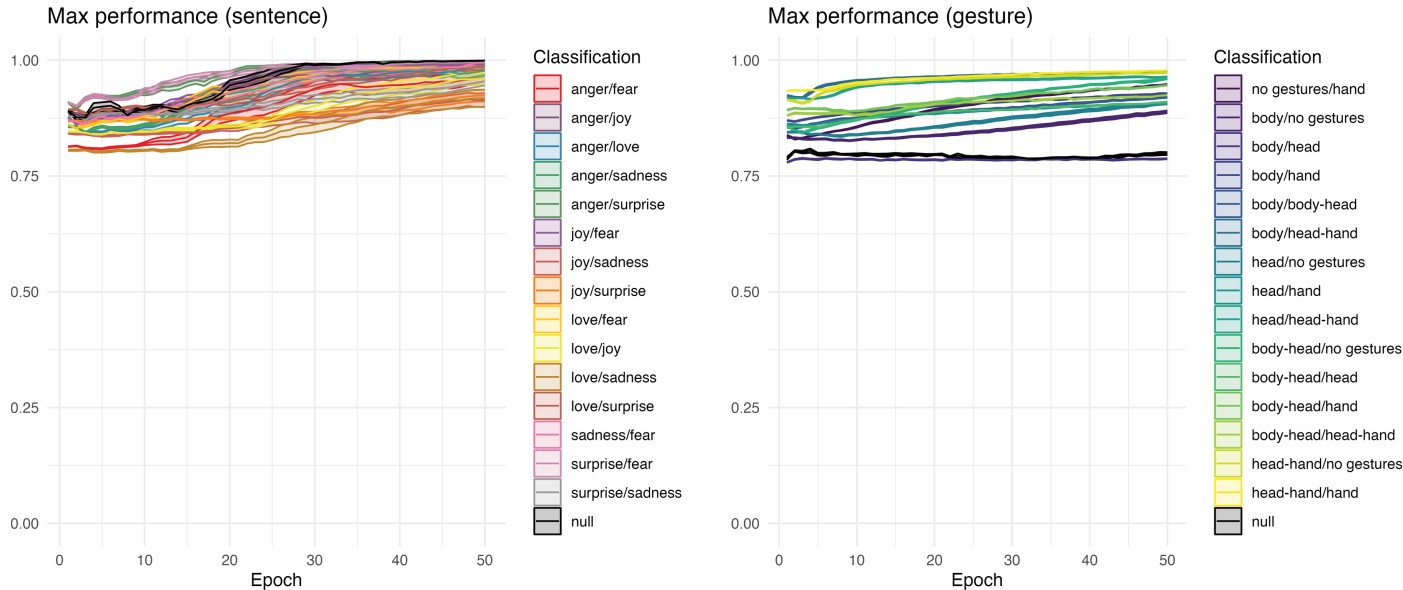

**Fig 7. *Max* performance for pairwise classifications.** Both sentence and gesture tasks show an increase in *max* performance over training epochs. On average this asymptotes near perfect performance but can vary depending on binary classification.

trial. Despite the small value, it is nevertheless significant and driven by the relatively higher performance in the sentence task.

The *max* performance also differs for both sentence ($R^2 = 0.0042$, $p < .0001$) and gesture tasks ($R^2 = 0.0722$, $p < .0001$) when the network learns from ordered datasets compared to randomly shuffled ones. The sentence conditions appear to pick up more spurious signals, ultimately approaching 1.00 as the epochs increase, while the gesture condition remains consistently unlearned across epochs. The learning pattern for $t_{max}$ across epochs (see next section) provides additional insights into these trends.

## Time at max

We would expect that completed training in the LSTM should show that maximum performance should appear near the *end* of an item, as this would suggest that the network has extracted useful information in the whole presentation. Indeed, this is indicated by lower time-at-max values earlier in the training. During the first few epochs, the maximum value occurs proportionally earlier in a training item, similar to the example shown above in Fig 5. However in both sentence and gesture datasets, networks slowly extract features across the stimulus items for classification, as shown in Fig 8. The time gradually rises for all classifications, though it can also vary widely and tends to be more irregular in gesture.

In a linear model predicting $t_{max}$ from data source, sentence and gesture are only slightly different, with 0.34% ($p < .0001$) of the variance accounted for. Classifications in sentence and gesture both relate significantly in a linear model predicting time at max, with sentence classifications accounting for 9.51% ($p < .0001$) of the variance and gesture classifications 24.95% ($p < .0001$).

Interpreting $t_{max}$ alongside *max*, we confirm that the learning experience of a language task is more reliant on spurious signals for reshuffled data compared to ordered data, as max continuously increases to reach 1.00 while $t_{max}$ remains sporadic. The gesture condition remains

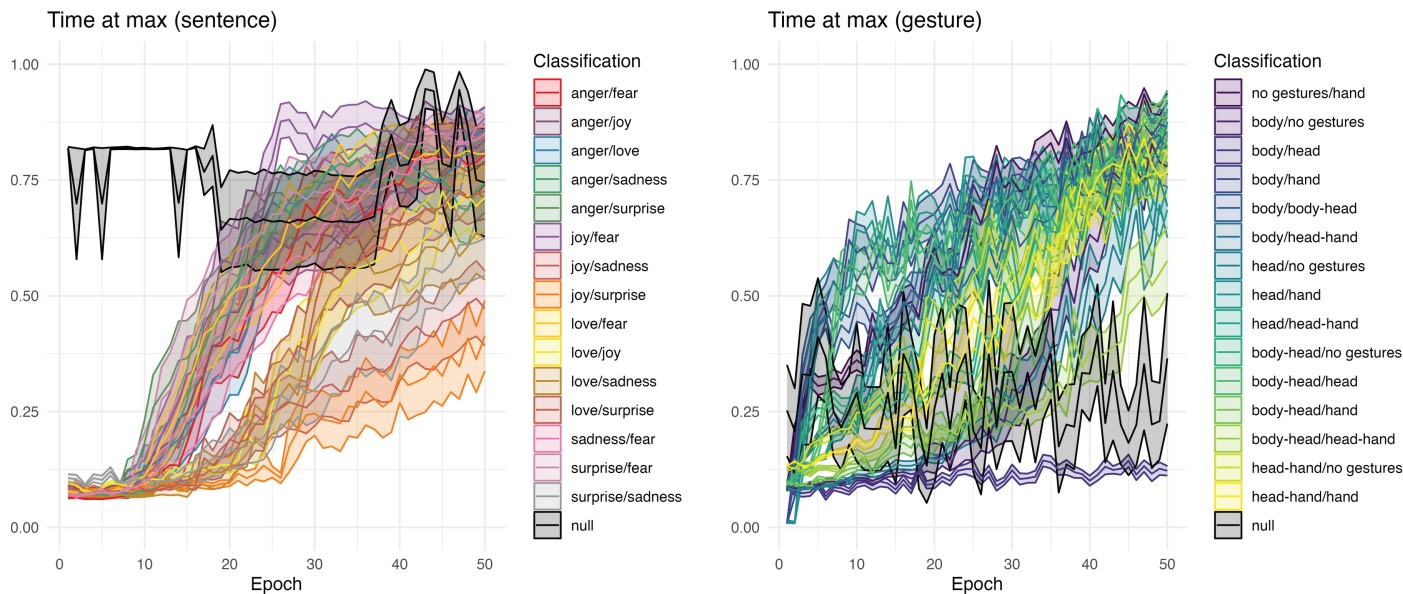

**Fig 8. Time at max ($t_{max}$) for pairwise classifications.** Both sentence and gesture tasks show an increase in the proportion of time at which maximum performance is observed. Indicative of information gain, this rises in both – suggesting the LSTM comes to better integrate the full test items for its performance. However this is much more orderly in the sentence task than the gesture task.

consistently unlearned across epochs for reshuffled data, as the *max* performance can occur at any point along the processing trajectory (any timestep). Both $t_{max}$ values indicate that no meaningful delayed signals were developed when the network was trained on superimposed reshuffled data. The time at max differs significantly for reshuffled versus ordered data in both sentence ($R^2 = 0.0120$, $p < .0001$) and gesture ($R^2 = 0.0058$, $p < .0001$) conditions.

## End - start performance

Finally, we wish to get a sense of how the LSTM models are extracting information within an information processing session. One way to do this is to discern how much higher its performance is at the end of a session compared to its performance initially. This is where we see the greatest difference between the datasets. As shown as Fig 9, sentence shows substantial information gain across stimulus items. By the end of training, the start and end of trials involves considerable difference, and the positive value of this difference shows that the network improves its performance significantly across presentations. With gesture, the situation is quite distinct. In most classifications, there is indeed a rise. But because gesture is informative even at the *first* time bin of an item, there isn't much information gain remaining. Indeed even the highest instances of this value hover near 0.10. Therefore, while performance on gesture is higher at first, the network may find the task more difficult in integrating that spatial information over time to improve performance.

As in the *start* measure, the difference between datasets is quite large. When data source is used to predict the *end - start* measure, 39.23% ($p < .0001$) of the observed variance in this measure can be associated with data source, with sentence associated with much higher information gain than gesture results. When looking at each data source separately, only 13.01% ($p < .0001$) of the measure is associated with the classification pairs in sentence. In the gesture

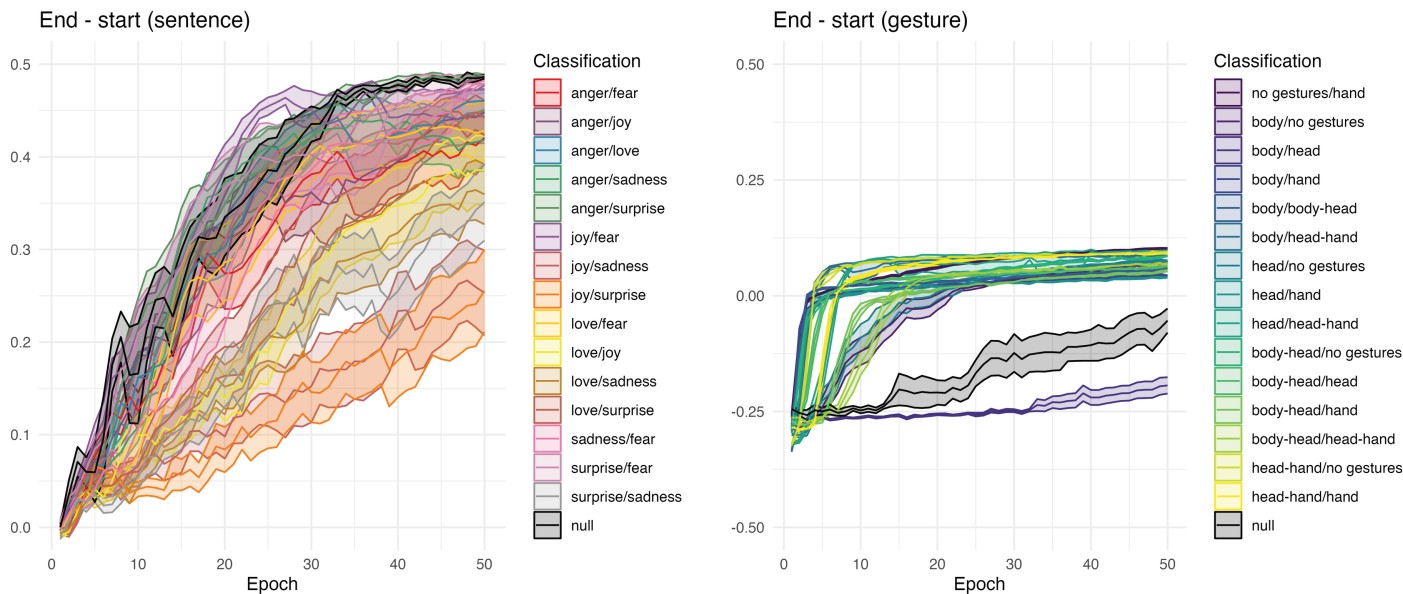

**Fig 9. *End - start* performance for pairwise classifications.** The information gain from start to end of stimulus item shows a stark difference between datasets. The sentence task shows that the LSTM's performance rises as it integrates data from start to end. With gestures, the story is more complicated, showing less improvement overall, and in one classification task, it is almost entirely unlearned.

dataset, this association is 38.08% ($p < .0001$), showing much higher contribution of the class differences in gesture.

Both sentence's ($R^2 = 0.0027$, $p < .0001$) and gesture's ($R^2 = 0.0340$, $p < .0001$) *end-start* values differ significantly between ordered and reshuffled data. Consistent with *max* performance gradually reaching 1.00, the information gain of the sentence network increases incrementally from 0.00 to 0.50 when trained on reshuffled data. Combined with insights from $t_{max}$, we suggest that although learning may appear to occur even with reshuffled data, it is likely driven by spurious signals picked up during the processing session. Additionally, the *end-start* performance for gesture gradually improved from highly negative to slightly negative but was not able to reach 0.00 during the early epochs (e.g. epochs smaller than 50), which explains the flat *max* performance for the reshuffled gesture condition.

## Discussion

Recognizing the importance and challenges involved in systematically unpacking the internal representations of DNNs, this study introduced a multi-dimensional quantification and visualization approach, "learning curves," which can capture two temporal dimensions of a model learning experience. First, it captures the "information processing trajectory," how the network is doing as it processes test items. Second, it captures the "developmental trajectory," describing how this processing is changing over training epochs. The former represents the influence of incoming signals on an agent's decision-making, which is operationalized by the timestep within a single epoch of an RNN. The latter conceptualizes the gradual improvement in an agent's decision-making abilities throughout its lifespan, operationalized by the iteration of epochs. The learning curve approach we illustrate in our two datasets shows that we can quantify and qualitatively investigate both of these dynamics within the same analysis, utilizing four descriptive metrics: *start*, *end - start*, *max*, $t_{max}$.

Based on a series of significant tests (see Table 2), we first demonstrated that there is a data source difference between sentence and gesture classification in the overall learning patterns across all four metrics (all $p$'s < .0001). Additionally, the learning experiences of both datasets' ordered classifications are distinct from their respective reshuffled null baselines, except for the *start* of the sentence classification. This is because both ordered (aggregated across all classes) and unordered binary sentence classifications will have an initial classification performance of 0.50 due to random guessing. This is corroborated by the significant test of the *start* metric for sentence class and gesture class: sentence pairwise classifications show very minor differences, with R-squared 0.10 ($p$ < .0001), whereas gesture classes show much greater divergence, with R-squared above 0.90 ($p$ < .0001), due to the initial spatial information contained in gesture coordinates, even at timestep 0, as mentioned in the previous results section. The significant difference between the reshuffled null baseline for ordered vs. unordered gesture classification is also due to different gesture class pairs having different prior spatial information, deviating from the 0.50 random guess scenario observed in sentence classification.

The modality distinction is further evident in the *end - start* metric (where sentence classification shows a much larger information gain than gesture classification, with gesture's information gain is more gradual). Finally, the ANOVA test capturing the R-squared difference between epoch × class versus epoch-only indicates significant distinctions between pairwise classes and not being distinguished by class (all $p$'s < .0001). This reveals that within the same multiclassification task, certain classes can be more difficult or easier to separate. Taken together, based on the analysis of these four measures derived from the learning curves across two distinct datasets, we highlight three insights gained from mapping these curves: *non-monotonicity*, *pairwise comparisons*, and *domain distinctions*.

## Non-monotonicity

First, we observed a non-monotonic trend in the learning experiences of DNNs. In other words, learning does not always show an increasing monotonic improvement because networks sometimes reveal temporary decrements in their performance across training. This is a characteristic recognized in previous literature as a key advantage for their advancement in addressing the most challenging tasks. For example, in applying neural network models to language, the presence of non-monotonic learning was taken as evidence that networks "reorganize" their knowledge as they learn. This may mean networks are acquiring more efficient representations of a problem space and the drop in performance is indicative of a transition into that more efficient representation. Classically, in the case of models learning language, they show a decrement in performance when they "discover a rule" in grammars [68] Our results are a testament to the observation that both representational consolidation and "catastrophic forgetting" remain as important issues in DNN learning [69].

Specifically in our results, we found that RNNs exhibit different preferences for early versus late cues when addressing various sequential tasks. For instance, in the sentence task, we noted a more pronounced performance improvement occurring in the later stages of information processing (i.e., model development). In contrast, gesture learning tends to show quicker progress, with more variability across epochs and repetitions of simulations, suggesting that DNNs tend to rely on shortcuts, such as naive cues related to keypoint coordinates, for gesture classification, rather than focusing on high-level movement sequences. This shortcut-based "learning" also is evident in the higher initial performance (*start*) for gesture classification (> 0.75). On the other hand, sentence classification begins at around 0.50 (the at-chance

probability) and exhibits greater performance gains in later epochs, indicating that the models classify based on high-level semantic sequences.

## Pairwise comparisons

Additionally, although multiclassification tends to exhibit collective model performance, our between-class pairwise comparisons reveal the presence of outliers within multiclassification. For example, "joy/fear", "surprise/fear", and "sadness/fear" demonstrate higher information gains across epochs compared to other emotion pairs, suggesting that these classes are further apart from each other. The "body/head" classification appears to experience learning challenges, possibly because these two movements have difficulty being completely separated, as the head's movement in naturalistic data may inevitably coincide with that of the body due to joint coordination.

The combination of learning curve mapping and instance measures thus serves as an effective approach for "auditing" representations in multiclassification problems. The proposed pipeline improves model explainability beyond holistic evaluation of classification performance and ad-hoc attention visualization by unpacking pairwise class learning patterns to reveal any pairs that are unsuccessful in being discriminated or involve delayed knowledge gain. This granular examination allows modelers to better investigate a model's appropriateness for the underlying task, as well as the properties of the processed input signals, reaffirming the value of our learning curve conceptualization.

## Domain distinctions

With these findings, it is tempting to infer that there are domain distinctions among different modality classification tasks. Gesture learning, for instance, may rely on autocorrelated signals (as body movements result from the coordinated contraction and relaxation of muscles), potentially emphasizing spatial semantics as early cues [3], while language learning relies on higher degrees of surprisal, irregularities and arbitrariness, as suggested by previous literature (see [43–46]). These domain distinctions are valuable to cognitive science in understanding how humans process and distinguish between various modes of communication, shedding light on the neural mechanisms underlying the flexibility and adaptability of the human mind when processing different forms of information and communication modalities.

Though it is intuitive and tempting to explain these domain distinctions here, we cannot yet assert that these trends would hold for *all* sentence or gesture (keypoint) classification tasks, only the ones we investigate here. However the learning curve results would seem to align with an intuition of how linguistic symbols would be sequentially integrated into a neural network in contrast to the highly auto-correlated spatial information contained in gesture performance. Still, we cannot infer a broad generalization about "language vs. gesture" and only leave it as a potential path for future investigation. Indeed, this may be an additional benefit of a method like the one we present here. Learning curves could finely ascertain these domain distinctions, and expanding the set of data to test may permit generalization in future work. This may have theoretical implications itself. The learning curve analysis may provide information about the distinct sources of information from varied modalities. When neural models (or human brains, presumably) integrate distinctive sources of information,

---

[3] The spatial information arising from the gesture, specifically the keypoint coordinates in the gesture dataset, can vaguely help the model distinguish between gesture classes (e.g., hand vs. no gesture). However, this information remains a low-level cue, as the model lacks an understanding of what a complete hand gesture sequence or a no-gesture sequence looks like.

they may strengthen understanding of complex multimodal data by leveraging their unique information-processing and developmental benefits.

## Limitations

This study provides conceptual and operational illustrations of applying learning curve methods to RNNs, though both the illustration and the proposed method come with certain considerations. First, we selected two distinct datasets to demonstrate how learning curves offer insights into different data and modality learning experiences. While illustrating significant domain distinctions, our study is limited in making broad generalizations between "language vs. behavioral." Future cognitive and behavioral research interested in such conceptual generalization can apply our proposed method to various distinct language tasks (e.g., language translation) and behavioral datasets (e.g., facial expression detection) to achieve broader generalizations across different modalities. Additionally, as highlighted at the beginning, our method could have implications for multimodal DNN modeling, given that our simulations utilize datasets with distinct modalities. For the current scope, we focus on how our proposed method facilitates the understanding of distinguishable modalities as an initial step, while future work can explore adopting this method for multimodal DNNs and datasets (e.g., audiovisual emotion classification) to assess how multimodal learning differs from unimodal learning.

From a methodological standpoint, as detailed in the model explainability technique comparison (Table 1), our proposed method is currently limited to temporal tasks and DNN architectures, specifically RNNs. However, it has the potential to be relatively easily adapted to various tasks, modalities, multiclassification, and different performance measures, beyond the binary classification used in this study. We have provided illustrations of the multiclassification learning curve based on sentence classification in S2 Text. Additionally, because our method generates measures at every timestep and epoch, it may be more computationally expensive compared to other explainability methods. However, this arises from the tradeoff between obtaining a more granular measure across all training steps versus only obtaining snapshot visualizations of a single neural network layer or attribution scores for all input features. As documented in the DNN explainability comparison (Table 1), each technique has its strengths and considerations, and researchers can exercise discretion in choosing the appropriate method. The learning curve method excels in providing a more systematic understanding of the model's learning process across models and modalities.

## Conclusion

There is a long tradition in cognitive science and computational neuroscience of examining the internal representations of models [9–12]. One reason for this is to determine if a model's features or processes reflect processes of the human mind. Such models can be informative for inferring properties of human mental processing, and so have direct theoretical implications. For example, Elman [13] and others [14] showed that RNNs can learn patterns sufficiently complex to resemble human grammar. By examining the internal activations of these recurrent networks, they showed that these systems are driven by graded, statistical features. Words are not discrete "symbols" but scalar vectors conditioned by linguistic context in time [15]. This was taken to challenge theories that see language as a purely abstract and symbolic recursive process [16].

Inspired by prior work, the learning curve method we proposed uses temporal mapping (information processing and developmental trajectories) to help modelers more comprehensively understand the underlying learning and decision-making processes of a complex model

architecture without delving into the intricacies of interpreting its internal representations directly [23,38]. This kind of systematic and quantitative approach has recently gained popularity in both computational cognitive science and deep learning communities [5,38,63–65], as it facilitates multidimensional comparisons across models and modalities, which were previously seen as challenging for DNN-like models. The current study illustrates multiple techniques for analyzing model learning experiences and highlights three insights across different communication modalities based on these analyses. Future studies can utilize this learning curve mapping approach to enhance model interpretability studies by evaluating a model's appropriateness for the task at hand, examining the properties of the underlying input signals, and assessing the model's alignment (or lack thereof) with human learning experiences, which is also a critical consideration for computational cognitive science and neuroscience research.

## Supporting information

**S1 Text.** Sentence and gesture RNN-LSTM model performance.
(PDF)

**Table A.** Average Performance metrics for sentence classification across all simulation runs (epochs = 50).
(PDF)

**Table B.** Average Performance metrics for gesture classification across all simulation runs (epochs = 50).
(PDF)

**Table C.** Average performance metrics for 10 examples of gesture classification across all simulation runs (epochs = 100).
(PDF)

**S2 Text.** Illustrations of multiclassification results for sentence classification (epochs = 20).
(PDF)

**S3 Text.** Illustrations of sentence and gesture simulations across 20 epochs.
(PDF)

**S4 Text.** Illustrations of gesture simulations across 100 epochs.
(PDF)

## Acknowledgments

We thank the organizers, panelists, and audience for allowing us to present the initial version of our work at the 53rd Annual Meeting of the Society for Computation in Psychology (SCiP) and for sponsoring the registration fee. We are also grateful to the reviewers from the 45th Annual Meeting of the Cognitive Science Society for their comprehensive review and invaluable feedback. Additionally, we thank Hongjing Lu, Jungseock Joo, and Elisa Kreiss for their insightful feedback on the theoretical framework and methodologies.

## Author contributions

**Conceptualization:** Yanru Jiang, Rick Dale.

**Data curation:** Yanru Jiang.

**Formal analysis:** Yanru Jiang, Rick Dale.

**Investigation:** Yanru Jiang, Rick Dale.

**Methodology:** Yanru Jiang, Rick Dale.

**Project administration:** Yanru Jiang, Rick Dale.

**Resources:** Yanru Jiang, Rick Dale.

**Software:** Yanru Jiang, Rick Dale.

**Supervision:** Yanru Jiang, Rick Dale.

**Validation:** Yanru Jiang, Rick Dale.

**Visualization:** Yanru Jiang, Rick Dale.

**Writing – original draft:** Yanru Jiang, Rick Dale.

**Writing – review & editing:** Yanru Jiang, Rick Dale.

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
