## [Decision Letter · Decision Letter 0]

19 Aug 2024

Dear MS Jiang,

Thank you very much for submitting your manuscript "Mapping the Learning Curves of Deep Learning Networks" for consideration at PLOS Computational Biology.

As with all papers reviewed by the journal, your manuscript was reviewed by members of the editorial board and by several independent reviewers. In light of the reviews (below this email), we would like to invite the resubmission of a significantly-revised version that takes into account the reviewers' comments.

We cannot make any decision about publication until we have seen the revised manuscript and your response to the reviewers' comments. Your revised manuscript is also likely to be sent to reviewers for further evaluation.

Sincerely,

Varun Dutt, Ph.D

Academic Editor

PLOS Computational Biology

Natalia Komarova

Section Editor

PLOS Computational Biology

Reviewer's Responses to Questions

**Comments to the Authors:**

Reviewer #1: See the attachment.

Reviewer #2: The manuscript titled “Mapping the Learning Curves of Deep Learning Networks” introduces a novel method for visualizing and quantifying the learning trajectories of deep neural networks. The study proposes a multidimensional approach that captures both the “information processing trajectory” and the “developmental trajectory” of a model’s learning process. By applying this method to tasks like gesture detection and natural language processing classification, the authors aim to provide insights into how models evolve over time and how their performance can be better understood. The manuscript discusses key findings such as nonlinearity in learning curves and the potential implications for cognitive processing and AI interpretability. Overall, while the manuscript is a suitable submission for PLOS Computational Biology, it requires minor revisions to enhance its scientific value.

1. First, the Abstract poorly outlines the purpose of the study, the problem statement, and the study's numerical results.

2. While nonlinearity is indeed a characteristic of deep neural networks, the manuscript does not provide a robust theoretical framework to justify why this nonlinearity is a key insight, nor does it connect this finding to broader principles in human cognitive processing or AI development.

3. The figures in the manuscript are presented as raster images of mediocre quality, which compromises their effectiveness in conveying critical information. High-quality vector images (e.g., SVG format) are essential for accurately depicting the nuances of the learning curves and the insights derived from them.

4. This research briefly touches on the application of the proposed method to multimodal tasks but does not provide a comprehensive analysis of its effectiveness in this area. It would benefit from a more detailed exploration of how the proposed learning curve analysis can be applied to multimodal tasks, with specific examples or simulations to demonstrate its effectiveness in such contexts.

5. The authors failed to explore the implications of temporal dynamics for different types of deep learning models, such as convolutional neural networks and transformers. The focus is primarily on recurrent neural networks, which limits the generalizability of the findings.

6. It would be much better, if the manuscript included a detailed comparison with other techniques, such as saliency maps, feature attribution methods, or layer-wise relevance propagation, to contextualize its contribution within the broader landscape of model interpretability. This might provide a clearer understanding of the method's strengths and weaknesses, thereby enhancing its scientific value.

7. The manuscript does not sufficiently address the limitations of the proposed method for mapping learning curves.

8. Finally, the Conclusion section is overly narrow and does not sufficiently summarize the study's numerical results and the proposed method's limitations.

In conclusion, the submitted manuscript "Mapping the Learning Curves of Deep Learning Networks" is a valuable contribution to the field and fits well within the scope of PLOS Computational Biology. However, minor revisions are required to address the issues outlined above. These revisions will enhance the manuscript's scientific rigor, practical value, and overall impact.

Reviewer #3: Dear Editor,

the manuscript "Mapping the Learning Curves of Deep Learning Networks" aims to contribute to the explainability of artificial deep neural networks in the context of machine learning (ML). Concerned that deep ML operates like a black box, the researchers propose four metrics ('start', 'max', 'tmax', 'end-start') to quantify a recurrent neural network training progression as a 'learning curve' in both steps and training epochs, which allows them to reason about 'nonlinearity', pairwise comparisons', and 'domain distinctions'. From a technical standpoint, the research deals with a classification task first, and applies original methodology to the internal states of the intermediate (training) steps. The reader is not provided with evidence of the quality of the final model. Unfortunately, the policy to stop at 20 iterations because 'the training performance shifted from underfitting to optimum to overfitting' is not technically sound. For example, the cited article by McLelland et al. used more than 2000 training iterations with an arguably simpler ML model. Absence of credible final output state undermines the empirical basis for the remaining arguments. As for statistical analysis, I would suggest to show significance with respect to a reasonable null hypothesis such as, for example, a random superposition of emotion or gesture states, instead of simply showing that all trials have a similar trend on a short training cycle. Conceptually, the computation of the metrics is quite intricate -- in what way do they add clarity to an opaque RNN model? A constructive analysis, perhaps by way of a simple example, could help understand this aspect better. Overall, the methodology could be both more concise and more detailed. I would recommend using a more structured approach and adhere to standard practices, e.g. architecture diagrams for neural networks.

The presentation uses a narrative style, requiring a considerable effort to follow and identify key original contributions. For example, a clear and concise statement about all four metrics being the main methodological contribution appears only on page 15. The research is not directly related to neuroscience or psychology research and as a result, large portions of introductory discussion is irrelevant for this work. Lastly, the authors are rightly concerned by over-reliance on AI for decision making, but wrongly attribute the fault to complicated models. The most common causes of ethical issues reside in flawed datasets, not complicated models (assuming sound technical execution and fair play).

Reviewer #4: Yes!

A PDF has been sent as an attachment.

**Have the authors made all data and (if applicable) computational code underlying the findings in their manuscript fully available?**

Reviewer #1: Yes

Reviewer #2: Yes

Reviewer #3: Yes

Reviewer #4: None

PLOS authors have the option to publish the peer review history of their article (what does this mean?). If published, this will include your full peer review and any attached files.

Reviewer #1: No

Reviewer #2: **Yes: **Dr. Pavlo Radiuk

Reviewer #3: No

Reviewer #4: No
---

## [Decision Letter · Decision Letter 1]

13 Jan 2025

Dear MS Jiang,

We are pleased to inform you that your manuscript 'Mapping the Learning Curves of Deep Learning Networks' has been provisionally accepted for publication in PLOS Computational Biology.

Best regards,

Varun Dutt, Ph.D

Academic Editor

PLOS Computational Biology

Natalia Komarova

Section Editor

PLOS Computational Biology

Reviewer's Responses to Questions

**Comments to the Authors:**

Reviewer #2: Dear Authors and Editor,

I am pleased to confirm that the authors have addressed all the critical comments and concerns raised in my previous review of the manuscript titled "Mapping the Learning Curves of Deep Learning Networks".

The authors have made significant revisions to the manuscript, including:

1. The abstract now provides a clear outline of the study's purpose, problem statement, and key numerical results, ensuring better accessibility for readers.

2. The authors have rephrased and clarified their observations on non-monotonic trends in learning metrics, ensuring theoretical consistency.

3. All figures have been converted to high-quality images, significantly improving their clarity and effectiveness in conveying critical insights.

4. The manuscript now includes detailed discussions on the applicability of the proposed method to multimodal tasks, with clear directions for future research.

5. The authors have justified their focus on RNNs while acknowledging the limitations and potential extensions to CNNs and transformers.

6. The addition of a comprehensive table comparing their proposed method with existing model interpretability techniques provides valuable context and highlights the unique contributions of their approach.

7. The authors have thoughtfully expanded the discussion of the method's limitations, adding balance and depth to the manuscript.

8. The conclusion now holistically summarizes the study's numerical results, contributions, and limitations, while engaging with broader implications for cognitive science and model interpretability research.

The revisions demonstrate the authors’ commitment to addressing the feedback comprehensively. They have effectively clarified ambiguities, rectified errors, and enhanced the manuscript's scientific rigor and presentation quality.

As a result, I now find the manuscript suitable for publication.

Sincerely,

Reviewer 2

Reviewer #4: About the article: POLCOMPBALL-D-24-01100.

The final report was sent as an attachment.

**Have the authors made all data and (if applicable) computational code underlying the findings in their manuscript fully available?**

Reviewer #2: Yes

Reviewer #4: Yes

PLOS authors have the option to publish the peer review history of their article (what does this mean?). If published, this will include your full peer review and any attached files.

Reviewer #2: **Yes: **Dr. Pavlo Radiuk

Reviewer #4: No

---

## [Editor Report · Acceptance letter]

PCOMPBIOL-D-24-01100R1

Mapping the Learning Curves of Deep Learning Networks

Dear Dr Jiang,

I am pleased to inform you that your manuscript has been formally accepted for publication in PLOS Computational Biology. Your manuscript is now with our production department and you will be notified of the publication date in due course.

With kind regards,

Anita Estes
